# Effects of biodegradable plastic film mulching on soil microbial communities in two agroecosystems

Sreejata Bandopadhyay[1], Henry Y. Sintim[2,3] and Jennifer M. DeBruyn[1]

[1] Department of Biosystems Engineering and Soil Science, University of Tennessee, Knoxville, TN, United States of America
[2] Department of Crop and Soil Sciences, Washington State University, Pullman, WA, United States of America
[3] Department of Crop and Soil Sciences, University of Georgia, Athens, GA, United States of America

## ABSTRACT

Plastic mulch films are used globally in crop production but incur considerable disposal and environmental pollution issues. Biodegradable plastic mulch films (BDMs), an alternative to polyethylene (PE)-based films, are designed to be tilled into the soil where they are expected to be mineralized to carbon dioxide, water and microbial biomass. However, insufficient research regarding the impacts of repeated soil incorporation of BDMs on soil microbial communities has partly contributed to limited adoption of BDMs. In this study, we evaluated the effects of BDM incorporation on soil microbial community structure and function over two years in two geographical locations: Knoxville, TN, and in Mount Vernon, WA, USA. Treatments included four plastic BDMs (three commercially available and one experimental film), a biodegradable cellulose paper mulch, a non-biodegradable PE mulch and a no mulch plot. Bacterial community structure determined using 16S rRNA gene amplicon sequencing revealed significant differences by location and season. Differences in bacterial communities by mulch treatment were not significant for any season in either location, except for Fall 2015 in WA where differences were observed between BDMs and no-mulch plots. Extracellular enzyme assays were used to characterize communities functionally, revealing significant differences by location and sampling season in both TN and WA but minimal differences between BDMs and PE treatments. Overall, BDMs had comparable influences on soil microbial communities to PE mulch films.

Corresponding author
Jennifer M. DeBruyn, jdebruyn@utk.edu

## INTRODUCTION

Plastic mulch films are widely used in crop production systems to improve soil microclimate and suppress weeds, translating into increased crop yields and/or improved fruit quality. Some of the agronomic benefits of using plastic mulch films include reduction of weed pressure (*Martín-Closas, Costa & Pelacho, 2017*), conservation of soil moisture (*Kader et al., 2017*; *Shahi et al., 2017*), and moderation of soil temperature. Low density polyethylene (PE) has been the favored polymer for mulch films due to its many attractive properties such as low cost, easy processability, high durability and flexibility (*Bandopadhyay et al., 2018*;

*Kasirajan & Ngouajio, 2012*). However, PE does not readily biodegrade, and thus must be disposed at the end of the growing season, contributing to our global plastic waste problem (*Brodhagen et al., 2015*; *Liu, He & Yan, 2014*). Even when removed from a field, fragments of film are left behind in the soil, which can affect soil function and soil biota (*Barnes et al., 2009*; *De Souza Machado et al., 2018b*; *Huerta Lwanga et al., 2016*; *Rillig, 2012*; *Sivan, 2011*; *Teuten et al., 2009*) or leach out into water systems and pollute aquatic ecosystems (*Fu & Du, 2011*; *Kong et al., 2012*; *Magdouli et al., 2013*; *Van Wezel et al., 2000*; *Wang et al., 2015*; *Wang et al., 2013*). As these plastics break down in soil, they form microplastics (*De Souza Machado et al., 2018a*), contributing to terrestrial microplastic pollution (*De Souza Machado et al., 2018a*; *De Souza Machado et al., 2018b*).

Plastic mulch use is expected to increase to meet increasing global food demands; therefore, it is imperative to find alternatives that will reduce the environmental footprint. Biodegradable mulch films (BDMs) are a potential alternative: BDMs are made of polymers that can be degraded by microbial action (*Hayes et al., 2012*; *Kasirajan & Ngouajio, 2012*; *Kyrikou & Briassoulis, 2007*; *Riggi, Santagata & Malinconico, 2011*). In the field, BDMs perform like other plastic films by altering the soil microclimate and improving crop yields (*DeVetter et al., 2017*). However, unlike PE plastics, which require removal and disposal, BDMs are designed to be tilled into the soil where resident soil microbes are expected to degrade them over time. Under ideal circumstances, BDMs should eventually be mineralized into carbon dioxide and water.

Despite being a promising sustainable alternative, adoption of BDMs has been limited in the US (*Goldberger et al., 2015*). BDMs are currently more expensive than PE mulches, and breakdown can be unpredictable. Growers and stakeholders have also cited insufficient knowledge regarding the effects of BDMs on soil health as a barrier to adoption (*Goldberger et al., 2015*). Moreover, the US National Organic Program (NOP) does not allow growers to use the currently available BDM products in organic crop production because they are not 100% biobased (*Miles et al., 2017*). However, the source of the carbon does not dictate biodegradability of BDMs; a BDM that is biodegradable and does not harm the soil, regardless of the source of feedstock, could become a sustainable alternative to PE mulch (*Ghimire et al., 2018a*). Thus, evaluating the impacts of incorporation of BDMs into soil on soil health is a critical part of adoption and policy development surrounding BDMs (*Brodhagen et al., 2017*).

BDMs can impact soil health in two ways: indirectly, in a manner similar to PE films, by acting as a surface barrier to soil and modifying the soil microclimate, and directly, by addition of physical fragments and carbon into soil after tillage (*Bandopadhyay et al., 2018*). The body of research on the impacts of polyethylene films on soil microbial communities and functions can help us predict the indirect effect of BDMs on soil health. However, research on the direct effects of BDMs on soil microbial community structure and function remains poorly answered due to a dearth of research that directly compares BDMs and PE in the same study. Unless there is a direct comparison of BDMs and PE, it is difficult to tease apart whether the observed changes are above and beyond what you would expect from the application of PE mulch to the soil surface (*Bandopadhyay et al., 2018*). These answers are critical if use of BDMs is to be advocated as a sustainable alternative to PE. Previous studies

have analyzed impacts of BDMs on soil microbial communities using phospholipid fatty acid (PLFA) profiling (*Li et al., 2014b*) and pyrosequencing (*Moore-Kucera et al., 2014*) methods. However, these studies did not use PE as a negative control so direct effects of BDMs on soil microbial community structure and function remain uncertain.

In this study, we compared the impacts of BDM and PE mulch on soil microbial communities using two-year vegetable crop field trials in two diverse climates (Knoxville, TN, in the southeastern USA and Mount Vernon, WA, in the northwestern USA). During this field trial, measurements of soil health indices based on a suite of soil physical, chemical and biological properties revealed that the overall effect of mulching on soil health indices was minimal and that BDMs performed comparably to PE (*Sintim et al., 2019*). The study by *Sintim et al. (2019)* included extracellular enzyme rates (expressed as C:N and C:P ratios), organic matter content and soil respiration as biological indicators of soil health. To build on this finding, we extended the study to focus on soil microbial communities, which are accepted as integral to soil functioning, but generally not explicitly included in assessments of soil health. Here, we evaluated the impacts of BDMs on: (1) soil microbial community structure, characterized using 16S rRNA gene amplicon sequencing; (2) soil microbial abundances, estimated using qPCR; and (3) soil microbial community function, estimated by a suite of soil extracellular enzyme rates over the two-year field trial experiment. We tested the hypothesis that plastic mulches would significantly alter soil microbial community structure and function, but that there would be no significant differences between PE and BDM mulches.

## MATERIALS & METHODS

### Plastic mulch films

Three commercially available biodegradable mulch films (BioAgri®, Naturecycle, and Organix A.G. Film™) and one experimental film comprised of a blend of polylactic acid (PLA) and polyhydroxyalkanoates (PHA) were tested alongside a polyethylene (PE) mulch (negative control), and cellulose paper mulch (WeedGuard Plus®, positive control). The paper mulch used in the experiment was a 100% biobased product and was chosen as the positive control because its major constituent is cellulose which is known to rapidly disintegrate in the field (*Li et al., 2014b*). Physicochemical properties of mulches are reported in Table 1.

### Field trial description

Field experiments were set up in two locations: East Tennessee Research and Education Center (ETREC), University of Tennessee, Knoxville, TN and the Northwestern Washington Research & Extension Center (NWREC), Washington State University, Mount Vernon, WA. The soil at Knoxville is a sandy loam (59.9% sand, 23.5% silt, and 16.6% clay), classified as a fine kaolinitic thermic Typic Paleudults. The soil at Mount Vernon is a silt loam (14.2% sand, 69.8% silt, and 16% clay), classified as a fine-silty mixed nonacid mesic Typic Fluvaquents. Henceforth in the paper, Knoxville, TN will be referred to as TN and Mount Vernon, WA will be referred to as WA. The mulches were tested in the field over two years (2015 to 2016) under pie pumpkin (*Cucurbita pepo*) as a test crop,

Table 1 **Manufacturers, major constituents, and physicochemical properties of the mulches used in the study.** Bio-based content was provided by the manufacturers. Data reported from *Hayes et al. (2017)*.

| Mulches | Manufacturer | Major constituents[a] | Weight ($g\ m^{-2}$) | Thickness ($\mu m$) | Elongation[b] (%) | Contact angle[c] (°) | Total carbon (%) | Biobased content (%) |
|---|---|---|---|---|---|---|---|---|
| BioAgri® | BioBag Americas, Inc., Dunedin, FL | Mater-Bi® grade EF04P (blend of starch and PBAT) | 18.0 | 26 | 260 | 87.6 | 57.6 | 20–25 |
| Naturecycle | Custom Bioplastics, Burlington, WA | Blend of starch and polyesters | 25.4 | 48 | 213 | 69.2 | 54.8 | ~20 |
| Organix A.G. Film™ | Organix Solutions, Maple Grove, MN | BASF®ecovio® grade M2351(blend of PLA and PBAT) | 17.8 | 20 | 273 | 86.2 | 51.4 | 10–20 |
| Experimental PLA/PHA | Metabolix Inc., Cambridge, MA | 88.4% MD05-1501 (56% Ingeo PLA, 24% Mirel™ amorphous PHA, 15% $CaCO_3$ and 5% additives), 10.0% Techmer PLA M91432 (20% carbon black in PLA 3052) and 1.6% PLA | 25.0 | 33 | 247 | 67.8 | 47.5 | 86 |
| WeedGuardPlus® | Sunshine Paper Co., Aurora, CO | Cellulose | 240 | 479 | 6.4 | <10 | 46.0 | 100 |
| Polyethylene | Filmtech, Allentown, PA | Linear low-density polyethylene | 25.4 | 47 | 578 | 79.3 | 82.9 | <1 |

**Notes.**
[a]PBAT, Polybutylene co-adipate co-terephthalate; PLA, Polylactic acid; PHA, Poly (hydroxyalkanoate).
[b]Measured in machine direction.
[c]Measured at 22 °C.

with full experimental details described in *Sintim et al. (2019)* and *Ghimire et al. (2018b)*. Briefly, before mulch application began in TN and WA, the plots were under winter wheat (*Triticum aestivum*) cover crop in TN and clover (*Trifolium spp.)* at WA. Each field site was arranged as a randomized complete block design with four replications of seven main plot treatments (six mulch treatments described above and one no mulch control). Mulches were machine-laid on raised beds at the end of May to early June, and harvest was completed in September-October. PE mulch was removed soon after harvest, while BDMs were tilled in; all beds were tilled within two weeks of harvest. The two sites were planted with a winter wheat cover crop following harvest in the fall. Pie pumpkin (*Cucurbita pepo*) was used as the test crop because it met requirements of a large-scale, multi-location field experiment; namely, pie pumpkin is economically important, commonly grown in both our field experimental locations, and has sufficient season length to maximize treatment exposure. Additionally, pie pumpkins, like other cucurbits such as cucumbers, melons and squash, are commonly grown on plastic mulch films throughout the United States (*Inglis, Miles & Wszelaki, 2015*).

Soil water content and temperature were monitored as described in *Sintim et al. (2019)*. Briefly, sensors (5TM, Decagon Devices Inc., Pullman, WA) installed in the center of each mulch treatment at 10-cm and 20-cm soil depths for one field block were connected to data loggers (EM50G, Decagon Devices Inc., Pullman, WA) that recorded the soil water and temperature data hourly. Soil water content and temperature data is reported in *Sintim et al. (2019)*. Air temperature, precipitation, relative humidity, wind, and solar radiation were collected from a meteorological station located at the field site at TN (Decagon Devices Inc. Weather Station, Pullman, WA), and about 100 m away from the field site at WA (WSU AgWeatherNet Station, Mount Vernon, WA). Weather data for the two locations were continuously collected from 2015–2017.

Soil physical, chemical, and biological properties were assessed over the two-year study for this site, in order to assess changes in soil health. Detailed protocols for these measurements and raw data is provided in *Sintim et al. (2019)*.

## Soil sampling

Soil samples were collected from each of the 28 plots (seven treatments, replicated four times) at both locations in the Spring (May) and Fall (September) of 2015 and 2016. Spring samples were collected approximately 2 weeks prior to mulch application. Fall soil samples were collected while the mulches were still in the field, approximately 2 to 3 weeks before the mulch was tilled in (BDMs) or removed (PE). In our studies, BDMs did not full degrade over the winter, so the 2016 samples represent communities that have been exposed to (and are presumably still degrading) tilled-in mulch from the previous season; Spring 2016 soil had been exposed to tilled-in mulches from the previous season, and Fall 2016 soil had been exposed to both tilled-in mulches from the previous season and new mulch laid for the 2016 season. Soil was collected from the top 10 cm using a 2 cm diameter stainless steel auger. Thirty 10-cm soil cores were taken about 20 cm apart and composited for each of the plots. All sampling equipment was cleaned with 70% ethanol between plots to limit cross contamination. Roots and pebbles were removed by hand, and soils homogenized

and stored in plastic bags for transport back to the lab. Soils were stored at −80 °C until DNA extraction and extracellular enzyme assays.

## Soil DNA extraction and quantification

Extraction of DNA from soil samples was completed using the MoBio™ PowerLyzer™ Power Soil DNA isolation kit (now branded under Qiagen™) with inhibitor removal technology, as per manufacturer's instructions. 0.25 grams of soil were used for the extractions, and the DNA obtained after the final elution step was stored at −20 °C until further analyses.

Quantification of the DNA extracted from soil was completed using the Quant-It™ PicoGreen™ dsDNA Quantification Kit (ThermoFisher Scientific) per manufacturer's instructions and quality of DNA was measured by 260/280 ratios in a NanoDrop™ Spectrophotometer (Table S1).

## Quantitative PCR for bacterial and fungal abundances

As a proxy for bacterial and fungal abundances, 16S rRNA (bacteria) and ITS (fungi) gene copy abundances were quantified from soil DNA samples using Femto™ Bacterial DNA quantification kit (Zymo Research) and Femto™ Fungal DNA quantification kit (Zymo Research) following the manufacturer's protocol. DNA extracts were diluted 1:10 prior to quantification and 1 μl of the diluted samples was used for each qPCR reaction. All samples were analyzed in triplicate. No-template negative controls were included in each run. Bacterial and fungal DNA standards were provided in the kit and the ng DNA standard per well was converted to copy numbers which were used for final calculations. qPCR reactions were performed in a CFX Connect Real-Time PCR Detection System (BioRad). qPCR efficiencies averaged around 85% and 90% for bacterial and fungal assays, respectively.

## DNA amplification and sequencing

16S rRNA amplicon sequencing of DNA extracts was conducted by the Genomic Services Laboratory (GSL) at Hudson Alpha, Huntsville, AL, following their standard operating procedures. Extracted DNA samples were shipped frozen in 96 well plates. The V4 region of the 16S rRNA gene was amplified using primers 515F (GTGCCAAGCAGCCGCGGTAA) and 806R (GGACTACHVGGGTWTCTAAT) (*Caporaso et al., 2012*). The first PCR used V4 amplicon primers, Kapa HiFi master mix, and 20 cycles of PCR. All aliquots and dilutions of the samples were completed using the Biomek liquid handler. PCR products were purified and stored at −20 °C until further processing was completed. PCR indexing was completed for 16S (V4) amplicons using GSL3.7/PE1 primers, Kapa HiFi master mix, and 12 cycles of PCR. Products were purified using magnetic beads using the Biomek liquid handler. Final libraries were quantified using Pico Green. V4 amplicon size obtained was 425 bp for the soil samples. The amplified 16S rRNA genes were sequenced using 250 paired-end reads on an Illumina MiSeq platform. Sequence reads were deposited in the NCBI sequence read archive (Accession PRJNA564156).

Raw sequence data was processed using mothur v.1.39.5 following the MiSeq SOP (*Schloss et al., 2009*). Before aligning to the reference database (SILVA release 102), unique

sequences were identified, and a count table generated. After alignment to SILVA database, sequences were filtered to remove overhangs at both ends, and sequences de-noised by pre-clustering sequences with up to two nucleotide differences. Chimeras were removed using the VSEARCH algorithm. Sequences were classified using the Bayesian classifier (*Wang et al., 2007*) against the Ribosomal Database Project training set (version 9) with a bootstrap value of >80% (*Wang et al., 2007*). Following this step, untargeted (i.e., non-bacterial) sequences classified as *Eukaryota* and *Arachaeota* were removed. Sequences were binned into phylotypes according to their taxonomic classification at the genus level. A consensus taxonomy for each OTU was generated by comparison to the RDP training set. The resulting OTU count table and taxonomy assignments were imported into R (v. 3.4.0) (*R Core Team, 2018*) for further downstream statistical analyses. Mothur code, R code and associated input files are available at: https://github.com/jdebruyn/BDM-Microbiology.

## Extracellular enzyme assays

Fluorescence microplate enzyme assays were conducted using fluorescently labelled substrates to assess enzyme activities in soil (*Bell et al., 2013*). Seven enzymes were assayed using their respective fluorescent substrates and standards (Table S2).

Soil slurries were prepared in a sodium acetate trihydrate buffer whose pH was matched closely with the soil pH. 800 μl of soil slurry was pipetted into deep well 96 well plates. Separate plates were prepared for 4-methylumbelliferone (MUB) and 7-amino-4-methylcoumarin (MUC) standard curves for each sample. 200 μl of appropriate standards and substrates were added to the soil slurries. The plates were sealed and inverted to mix the contents. Incubation was done for 3 h at room temperature, after which the substrate and standard plates were centrifuged at 1,500 rpm (~327 × g) for 3 min. The supernatants were pipetted into black 96 well plates and fluorescence measured at 365 nm excitation wavelength and 450 nm emission wavelength in a BioTek® Synergy plate reader.

## Statistical analyses

Beta diversity was computed using Bray-Curtis distances of microbial community composition using the vegan package (v 2.4-3) in R version 3.4.0 (*R Core Team, 2018*) based on OTU tables, and were then visualized using non-metric multidimensional scaling (NMDS) using phyloseq package v.1.21.0 in R (*McMurdie & Holmes, 2013*). To determine whether significant differences existed in bacterial community composition between bacterial communities across different locations, seasons, and mulch treatments, a permutational multivariate analysis of variance (PERMANOVA) was performed using the ADONIS function implemented in R, based on the Bray–Curtis dissimilarity matrix. All libraries were subsampled to even depth (minimum sample read count, i.e., smallest library size, of 34,266) before analysis was performed. Similarity percentage analyses (SIMPER) was completed in R to reveal the most influential OTUs driving differences between soil bacterial communities in different locations, and across different seasons. Differences in relative abundances of taxa between locations and seasons were determined using Kruskal-Wallis rank sum non-parametric test. A post-hoc test was completed using pairwise Wilcoxon

rank sum test if significant differences were reported using Kruskal-Wallis test. *P* values were adjusted using the method of *Benjamini & Hochberg (1995)* to control the false discovery rates ($p < 0.05$). Canonical analysis of principal coordinates (CAP) was done to relate environmental variables reported in *Sintim et al. (2019)* to changes in bacterial community composition. Our *a priori* hypothesis for conducting this statistical test was that we would see changes in microbial communities across the two locations driven by specific environmental variables. The ordination axes were constrained to linear combinations of environmental variables, then the environmental scores were plotted onto the ordination. A PERMANOVA was performed on the CAP axes. These analyses were completed in R following the online tutorial by *Berry (2016)*.

Libraries were subsampled with replacement to equal size prior to computing alpha diversity metrics. The estimate_richness function was used in R phyloseq package to calculate observed richness and inverse Simpson indices (for diversity). A mixed model analysis of variance was completed using the generalized linear mixed model (GLIMMIX) procedure in SAS V. 9.3 to assess changes in richness and inverse Simpson over time. The fixed effects were location (TN and WA), mulch treatments (seven treatments) and time/season of soil sampling (four time points), while the random effect was block (three replicates). Repeated measures were incorporated in the model as sampling was done over time. The model was a completely randomized design (CRD) split-split-plot with repeated measures in the sub-sub plot. Normality of data was checked using Shapiro–Wilk test ($W > 0.9$) and equal variance using Levene's test ($\alpha = 0.05$). All data were normal and hence no transformations were performed.

To visualize differences in the functional profile of the communities; i.e., all seven enzyme rates, NMDS ordination of Bray-Curtis similarities was done in Primer 7 v. 7.0.13 (PRIMER-E). A mixed model analysis of variance with repeated measures was completed using the generalized linear mixed model (GLIMMIX) procedure in SAS V. 9.3 to assess changes in enzyme activities over time. Fixed and random effects were same as specified above. Boxplots for equal variance and outliers, reported in SAS, were used to remove outliers in the dataset. Normality was checked using Shapiro–Wilk test ($W > 0.9$) and probability plots for residuals, and equal variance using Levene's test ($\alpha = 0.05$). Data were log transformed as necessary when these conditions were not met. Type III tests of fixed effects and interaction effects are reported.

To assess for potential enrichment of bacteria and fungi, a paired $t$-test was conducted using initial and final 16S rRNA and ITS gene copy abundances (determined by qPCR) from Spring 2015 and Fall 2016 to see if there was a significant change over the course of the experiment. Initial 16S and ITS gene copy abundances from Spring 2015 were also subtracted from final abundances in Fall 2016 to get change in abundance over time. To determine if the enrichment or depletion of bacterial and fungal abundances was significantly different between treatments, a mixed model analysis of variance in SAS v. 9.3 using the GLIMMIX procedure was conducted on the differences. Significance level of all analyses were assessed at $\alpha = 0.05$. All data were checked for normality using Shapiro–Wilk test ($W > 0.9$).

# RESULTS

## Environmental and soil physicochemical data

Environmental data collected during the experiment is reported in *Sintim et al. (2019)* and in Table S3. The mean daily air temperature in 2015 to 2016 was about 4 °C higher in Knoxville, TN than in Mount Vernon, WA (Table S3). The total annual precipitation during the experimental years was higher in Knoxville, TN than in Mount Vernon, WA.

In all plots with plastic mulching, fragments of the mulches (i.e., remnants from the previous season's mulches) were visible in the soils throughout the experiment. Soil temperature, moisture and physicochemical properties were measured and reported previously by *Sintim et al. (2019)*. In summary, significantly increased soil temperature was observed in the early growing seasons in the plastic mulch plots compared to the cellulose and no-mulch plots. On average, the monthly soil temperature was greater in TN than in WA. Overall, mulched plots had higher water content than the no mulch plots, with PE mulch having the highest soil water content for the greatest time. The soil health analysis revealed some effects of mulching on certain properties (namely aggregate stability, infiltration, soil pH, electrical conductivity, nitrate, and exchangeable potassium), but these were not consistent among BDMs, nor across sampling times and locations.

## Soil bacterial community diversity and structure

For the 16S rRNA gene sequences, the percentage of bases with a Phred quality (Q) score ≥30 was 78% (24,090,356 total reads with 94% reads identified), and 90% (21,712,542 total reads with 93% reads identified) for the two flow cells used. The NMDS ordination revealed a clear difference in community structure between TN and WA when combining data from all four sampling seasons (Spring 2015 to Fall 2016) (Fig. 1A). Permutational ANOVA (PERMANOVA) tests confirmed significant differences between TN and WA soil microbial communities (Table 2, Table S4). The mean relative abundances of the most abundant classes of bacteria are shown in Fig. 1B. Similarity percentage tests (SIMPER) revealed the most influential OTUs contributing to the variation seen between location (Fig. 1B). The most influential OTUs belonged to several classes of microbes such as *Acidobacteria_Gp7, Acidobacteria_Gp16, Acidobacteria_Gp4, Planctomycetacia* and *Spartobacteria*. CAP analysis revealed that the differences in soil communities between TN and WA were most related soil moisture and organic matter content (Fig. S1).

In addition to locational differences, bacterial communities also differed significantly between the different seasons (Table 2, Table S4). For both locations, Spring communities were more similar to each other than Fall communities (Figs. 2A and 2B). SIMPER tests revealed that several genera of *Acidobacteria, Planctomycetaceae, Spartobacteria* and *Actinobacteria* (such as *Streptomyces*) were cumulatively responsible for 60% of the seasonal variance in bacterial communities (Figs. S2 and S3). Interestingly, relative abundance of *Streptomyces* increased over time from Spring 2015 to Fall 2016 in both TN and WA (Fig. S2).

Unlike location and season, the mulch treatments did not have a significant effect on bacterial community structure. Because of the locational and seasonal differences, we additionally analyzed each time-location set separately, and did not detect any significant

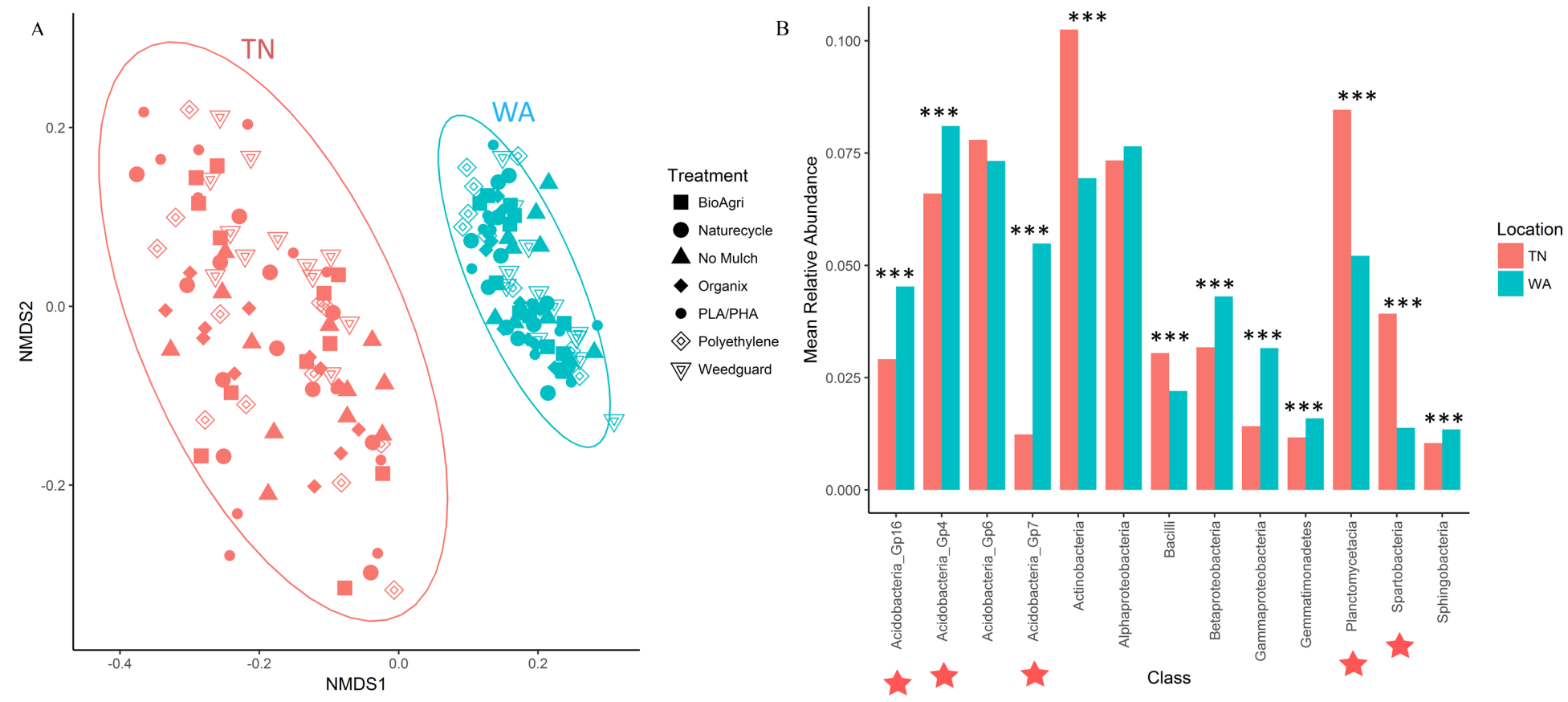

**Figure 1 Bacterial community composition differences between the two field locations, showing communities from all four sampling times.** (A) Non-metric multidimensional scaling (NMDS) ordination of Bray-Curtis dissimilarities of OTU relative abundances, highlighting differences between location (PERMANOVA $p = 0.001$). Each point corresponds to the microbial community of one plot in the field (4 sampling times * 3 replicate plots, resulting in 12 points for each treatment). Ellipses denote clustering at 95% confidence. NMDS stress value: 0.14. (B) Bar plot showing differences in mean relative abundance of the most abundant classes of bacteria in TN and WA, aggregating all treatments and all four sampling times. Asterices denote significant differences between locations, determined by ANOVA ($^*p \leq 0.05$, $^{**}p \leq 0.01$, $^{***}p \leq 0.001$). Red stars indicate taxa which cumulatively contributed up to 46% of the variance in microbial communities between TN and WA, determined using SIMPER.

**Table 2 PERMANOVA results (*F* values) showing differences in bacterial community composition by location, time and mulch treatment.** Significant differences are in bold; $^*p < 0.05$; $^{**}p < 0.01$; $^{***}p < 0.001$.

| Factor/treatment | Levels | TN (F) | WA (F) |
|---|---|---|---|
| Location | TN, WA | **117.34*****（TN and WA） | |
| Time | Spring 2015, Fall 2015, Spring 2016, Fall 2016 | **17.83***** | **32.84***** |
| Mulch treatments (Spring 2015—initial sampling) | 7 treatments: 5 BDMs[a] (BioAgri, Organix, PLA/PHA, Naturecycle, Weedguard), PE[b], no mulch control | 0.61 | 0.81 |
| Mulch treatments (Fall 2015) | | 0.87 | **1.96**** |
| Mulch treatments (Spring 2016) | | 0.84 | 0.81 |
| Mulch treatments (Fall 2016) | | 1.15 | 1.26 |

**Notes.**
[a]BDMs, biodegradable mulches
[b]PE, polyethylene

effects of treatment on community structure except for Fall 2015 in WA (Fig. S4, Table 2, Table S4). We further used a pairwise.adonis function in R (*Salazar, 2019*) to determine pairwise differences for Fall 2015 in WA, but no significant patterns emerged.

Alpha diversity of the soil bacterial communities was estimated using observed species richness and inverse Simpson index of diversity (Table S5). The observed species richness estimator measures count of unique OTUs in each sample. There were significant differences between TN and WA ($p < 0.05$) in richness estimates (Table 3, Fig. 3A). TN had greater richness than WA throughout the experiment, ranging from 260 to 300 unique OTUs. WA richness estimates ranged from 250 to 280 OTUs over the two years.

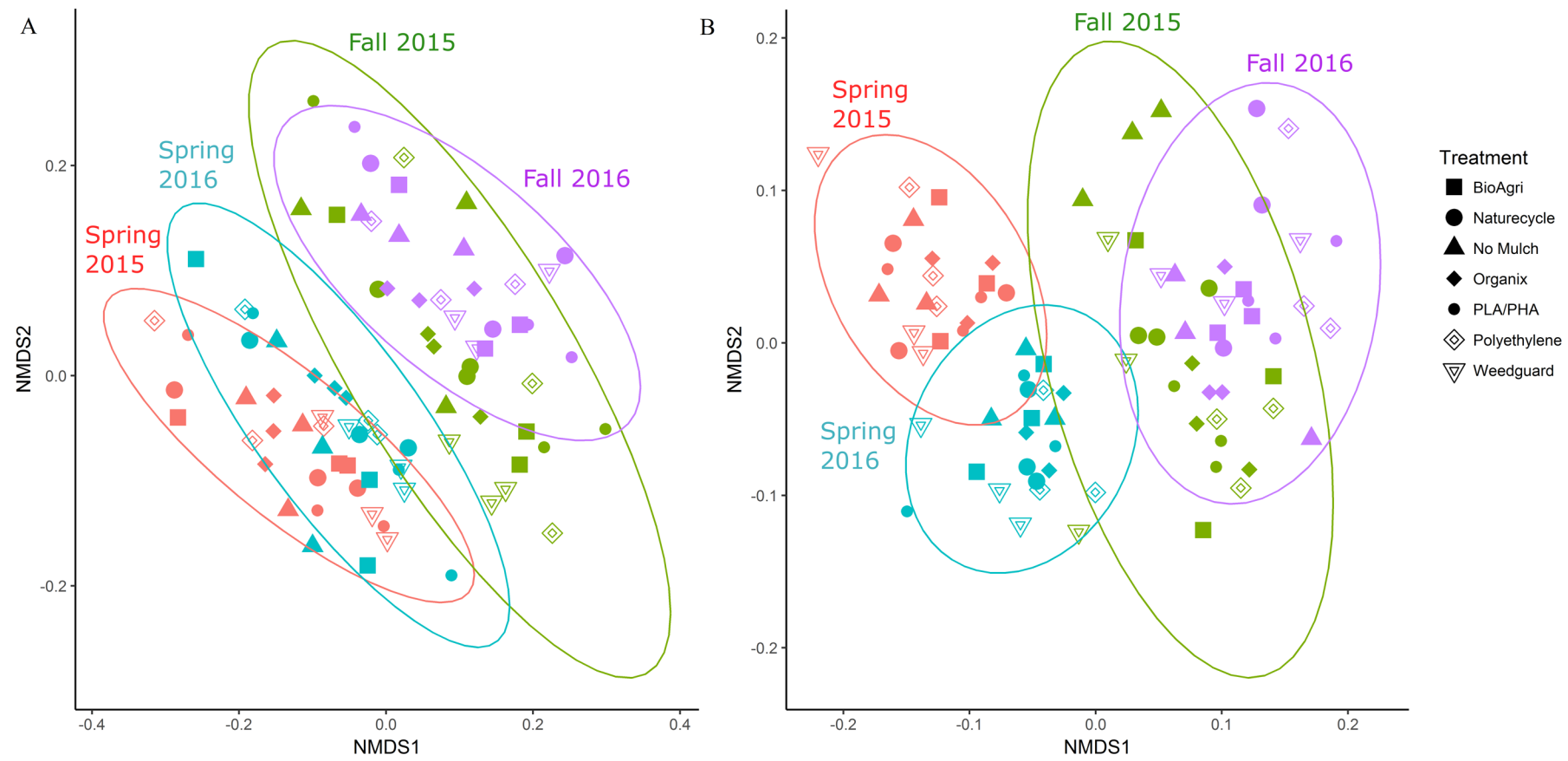

**Figure 2 NMDS ordination of Bray Curtis dissimilarities of soil bacterial communities showing significant differences between season.** (A) TN (PERMANOVA $p = 0.001$; stress: 0.17) and (B) WA ( $p = 0.001$; stress: 0.16). Ellipses denote clustering at 95% confidence. Spring 2015 samples represent initial soils prior to mulch application. Fall 2015 represents soils exposed to a season of surface applied mulches. Spring 2016 and Fall 2016 represent soils exposed to BDM mulch fragments tilled into soil from the previous season.

**Table 3 *F* values obtained from a mixed model analysis of variance of the alpha diversity metrics richness (number of observed OTUs) and diversity index (inverse Simpson).** Significant values are in bold, $*p < 0.05$; $**p < 0.01$; $***p < 0.001$.

| Factor/treatment | Levels | Richness F | Diversity F |
|---|---|---|---|
| Location | TN, WA | **24.42*** ** | 2.98 |
| Treatment | 7 treatments: 5 BDMs[a] (BioAgri, Organix, PLA/PHA, Naturecycle, Weedguard), PE[b], no mulch control | 1.93 | 1.20 |
| Location*Treatment | | 1.22 | 1.58 |
| Time | Spring 2015, Fall 2015, Spring 2016, Fall 2016 | **19.28*** ** | **122.23*** ** |
| Location*Time | | **6.06*** ** | **3.84** ** |
| Treatment*Time | | **2.4** ** | 1.63 |
| Location*Treatment*Time | | 0.55 | 1.09 |

**Notes.**
[a]BDMs, biodegradable mulches.
[b]PE, polyethylene.

The locational differences in richness were due to a lower richness in Fall 2015, Spring 2016 and Fall 2016 in WA (Fig. 3A). The inverse Simpson diversity index ranges were similar between TN and WA, ranging from 7 to 11 (Fig. 3B).

For both TN and WA, there was a significant difference between seasons in both richness and inverse Simpson index (Table 3). In TN in 2016, PE had the lowest richness and BioAgri had the highest. However, treatment differences in richness estimates were not significant (Table 3) when analyzing data using a mixed model. Inverse Simpson diversity indices were also not significantly different between treatments (Table 3). Looking at the final time point in TN, diversity estimates were highest for Weedguard, and lowest for PE; in WA, the estimates were highest for Weedguard, followed by PE with BDMs having lower diversity than PE or Weedguard; however, these differences were not significant (Fig. 3B).

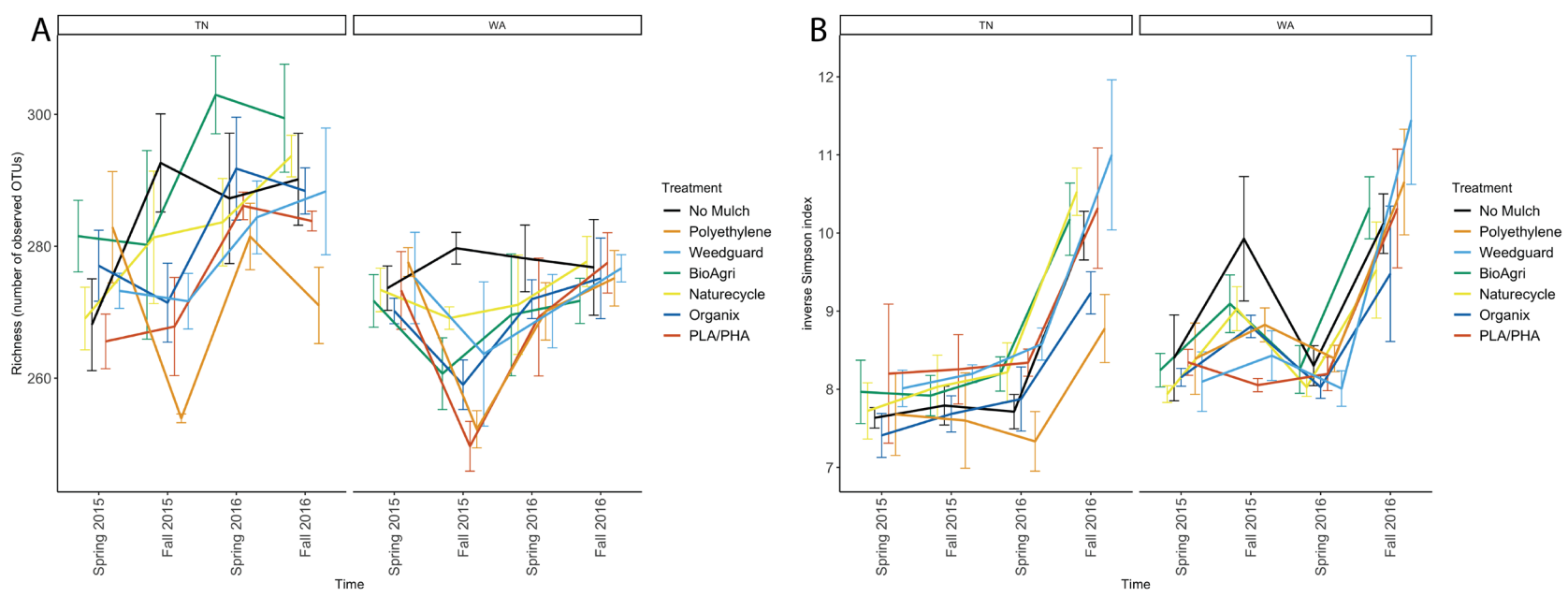

**Figure 3 Changes in alpha diversity of soil microbial communities over time in TN and WA.** (A) Richness (number of unique OTUs) and (B) inverse Simpson estimates. Spring 2015 samples represent initial soils prior to mulch application. Error bars indicate SEM of three replicate samples.

## Microbial community abundances

As a proxy for bacterial and fungal abundances, bacterial (16S) and fungal (ITS) rRNA gene copies were quantified using qPCR assays for soil samples from all seasons. In general, bacterial (16S) gene copy numbers averaged to $2.5 \times 10^9$ gene copies per gram dry weight soil in Spring 2015 and $4.2 \times 10^9$ gene copies $g^{-1}$ in Fall 2016 in TN; $1.8 \times 10^9$ gene copies $g^{-1}$ in Spring 2015 and $6.9 \times 10^9$ gene copies $g^{-1}$ in Fall 2016 in WA. For fungal (ITS) abundances, values ranged from $2.9 \times 10^8$ gene copies $g^{-1}$ in Spring 2015 to $3.8 \times 10^8$ gene copies $g^{-1}$ in Fall 2016 in TN, and $4.5 \times 10^8$ gene copies $g^{-1}$ in Spring 2015 to $8.4 \times 10^8$ gene copies $g^{-1}$ in Fall 2016 in WA. In order to assess if gene abundances had significantly changed over the course of the experiment (Spring 2015 to Fall 2016) for each mulch treatment, a paired $t$-test was used to identify changes that are significantly different from zero (Table S6). There was a significant increase in bacterial gene copies under BDM and Weedguard treatments in WA, but no significant change for no mulch and PE treatments (Table S6). There was also a significant enrichment in fungal gene copies for two of the BDMs (PLA/PHA and Naturecycle) in WA. In TN, significant enrichment in bacterial gene copies was seen under Organix, PLA/PHA and PE treatments (Table S6) but no enrichment was seen in fungal gene copies. In order to determine if these changes were significantly different between treatments, the differences between the final (Fall 2016) and the initial (Spring 2015) abundances were analyzed using a mixed model analysis of variance and Tukey post hoc tests. In both locations, mulch treatments did not have a significant effect on the changes in either 16S or ITS gene copies over the course of the experiment (Figs. 4A and 4B).

## Microbial community functions

To assess potential functional responses of the soil microbial communities to the mulching treatments, extracellular enzyme potential rate assays were conducted for common carbon, nitrogen, and phosphorus cycling enzymes in soil (Table S2). The data were combined over the two years to visualize Bray Curtis similarities of the enzyme rate profiles (Fig. 5). Locational differences in the enzyme profile were significant ($p < 0.05$), as were seasonal differences in both TN ($p < 0.05$) and WA ($p < 0.05$) evaluated using PERMANOVA

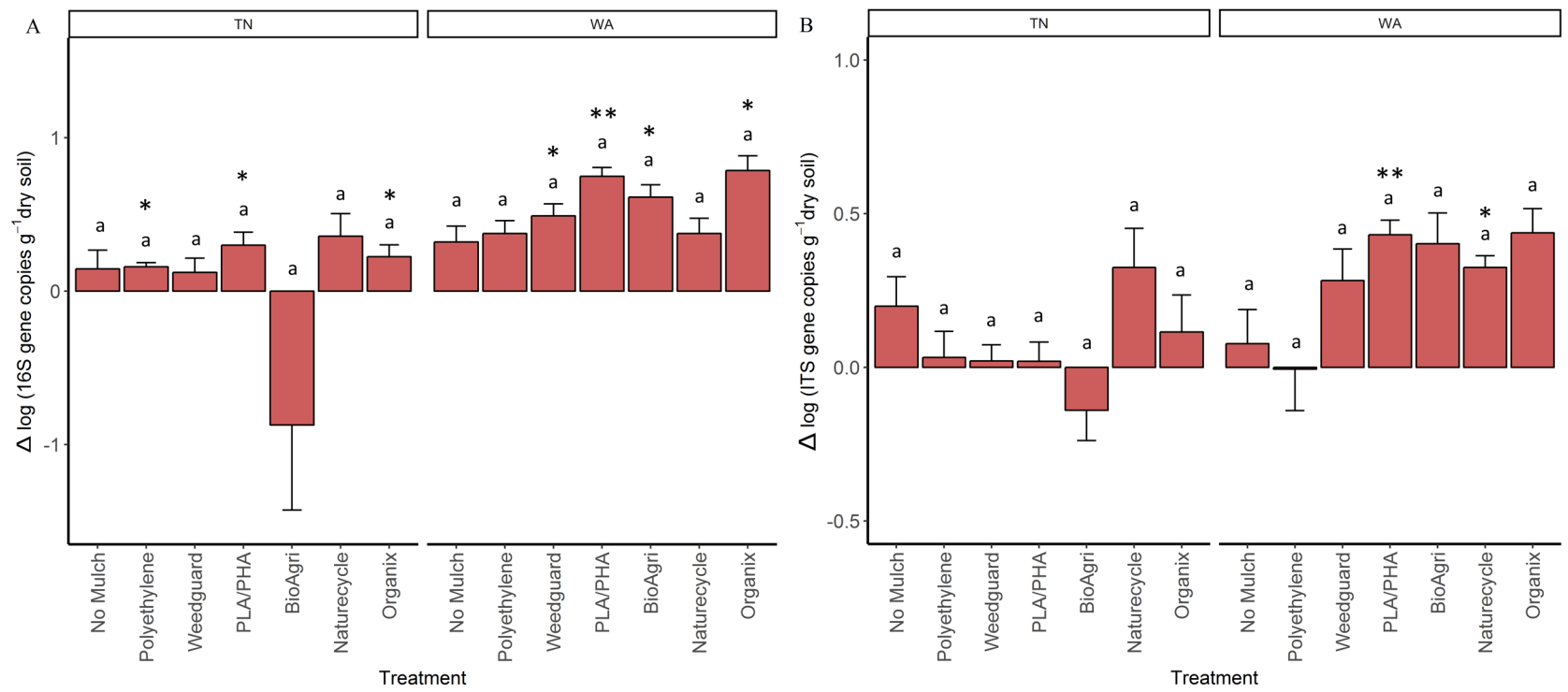

**Figure 4** **Change in bacterial and fungal abundances over the duration of the two-year experiment in TN and WA.** Net change in (A) bacterial 16S rRNA and (B) fungal ITS gene copy abundances per gram dry weight soil. Net changes were calculated by subtracting starting abundances (Spring 2015) from final abundances (Fall 2016). Error bars are SE of four replicate samples. Lowercase letters denote significant groupings between treatments ($p \leq 0.05$, Tukey's HSD). Asterices indicate treatments which showed significant enrichment (i.e., significant difference from 0) using a paired $t$-test (*$p \leq 0.05$, **$p \leq 0.01$, ***$p \leq 0.001$).

(Fig. 5). However, mulch treatment did not have a significant effect on the enzyme profile for any of the seasons at either location ($p < 0.05$). NMDS ordination for the final sampling time point Spring 2017 is shown in Fig. S5, showing no clear treatment differences. In general, the enzyme activity rates oscillated between higher activities in the Spring and lower activities in the Fall. When analyzed separately for each enzyme, the data over the two years revealed a significant effect of sampling time in TN for all seven enzymes assayed. In WA, enzyme activities of β-xylosidase, β-glucosidase, α-glucosidase, N-acetyl β-glucosaminidase and phosphatase were significantly different between sampling times (Fig. 6). In WA, cellobiosidase and leucine amino peptidase activities remained unchanged across the seasons (10–22 nmol activity $g^{-1}$ dry soil $h^{-1}$ for cellobiosidase and 200–375 nmol activity $g^{-1}$ dry soil $h^{-1}$ for leucine amino peptidase) (Fig. 6).

When averaged across seasons, mulch treatment differences were not significant for any soil enzymes in WA (Table 4). However, in TN, an effect of mulch treatment was observed for N-acetylβ-glucosaminidase activities (Table 4). N-acetyl β-glucosaminidase activity was reduced under BDMs and PE compared to no mulch plots. Interaction effects of mulch treatment and time of sampling were not detectable for any of the enzymes assayed in TN or WA (Table 4).

## DISCUSSION

In this study, soil microbial community composition was not significantly altered by mulch type. This is in contrast to other studies that have reported altered bacterial communities in soils under BDMs (*Koitabashi et al., 2012*; *Li et al., 2014b*; *Muroi et al., 2016*), and under non-biodegradable plastic mulches (*Farmer et al., 2017*; *Munoz et al., 2015*). Such opposite findings could be due to differences in methodology: For example, the studies by *Koitabashi et al. (2012)* and *Muroi et al. (2016)* were shorter laboratory

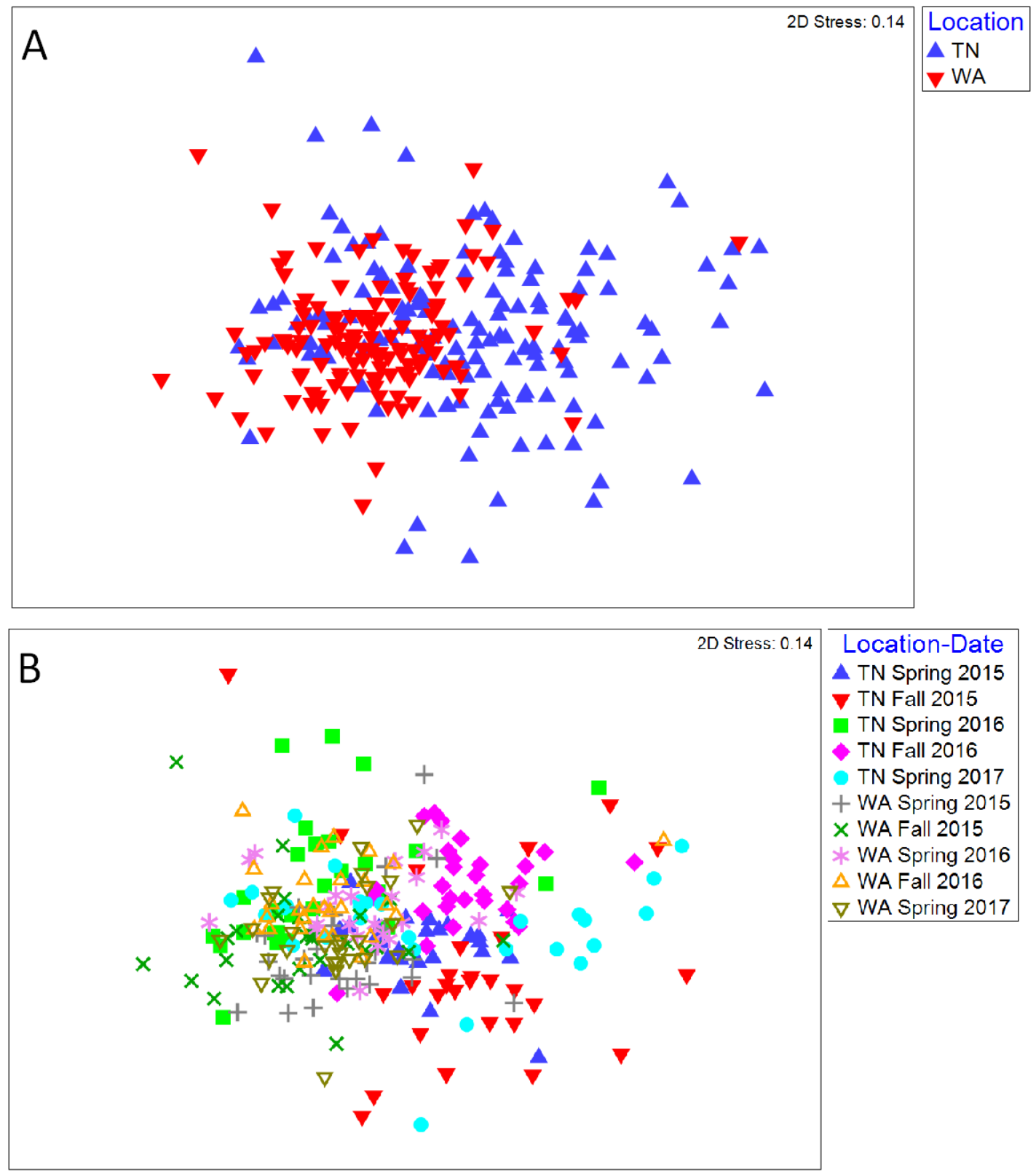

**Figure 5 NMDS ordination of Bray-Curtis similarity of the functional profile of soil microbial communities (based on 7 soil enzyme activities).** Points are differentiated by (A) location and (B) sample time. Spring 2015 samples represent initial soils prior to mulch application.

incubation studies in controlled conditions (28 °C to 30 °C), used pure polymer feedstock rather than commercial film formulations which include plasticizers and other additives, and relied on different detection methods such as polymerase chain reaction-denaturing gradient gel electrophoresis (PCR-DGGE). Laboratory studies under controlled conditions often result in more rapid microbial responses to treatments compared to field studies where variable environments introduce more noise. Our lack of observed difference may also be because we used a realistic, but low, plastic to soil ratio: For example, in the study by *Muroi et al. (2016)*, 1.8 g PBAT films was used in 300 g soil. In the field, tilled-in BDMs are a very small input of carbon when taking into account the volume of soil into which they are incorporated (*Bandopadhyay et al., 2018*). For comparison, the input of mulch carbon added to the soil in this study was a significantly smaller amount (6 to 25 g C $m^{-2}$) (*Hayes et al., 2017*) compared to the amount typically added from cover crop residues (142 g C $m^{-2}$) (*Al-Kaisi & Lal, 2017*). However, several studies have demonstrated responses by soil microbes to these small inputs (*Bandopadhyay et al., 2018*), suggesting that even if they are not a major carbon source, they do influence microbial activities by some other mechanism, and may result in a difference between BDMs and PE after multiple seasons of BDM incorporation. Finally, because our aim was to characterize responses in bulk soil

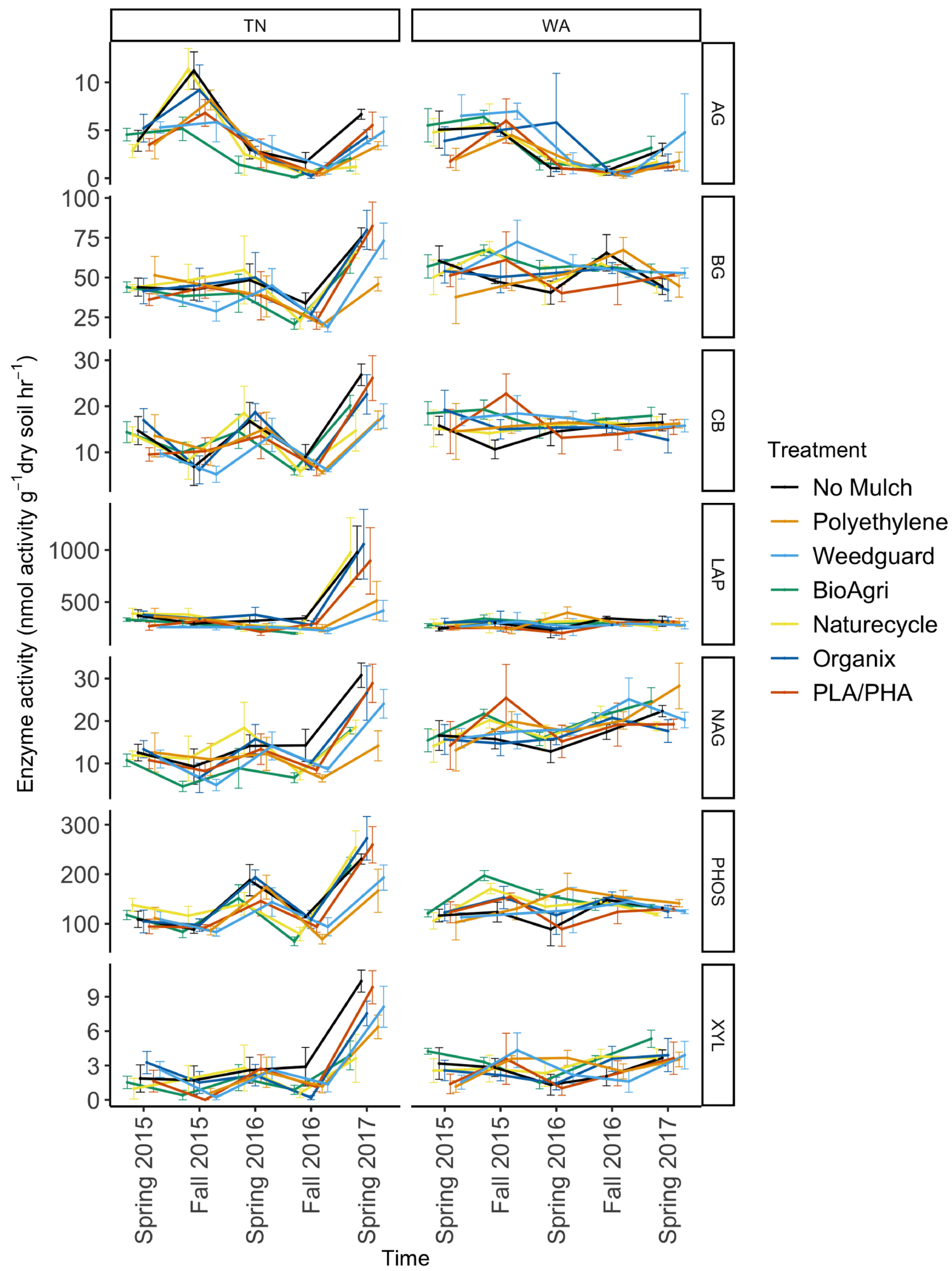

**Figure 6 Soil enzyme activity over time in TN and WA under different mulching treatments.** *P* values are reported in Table 4. Spring 2015 samples represent initial soils prior to mulch application. Error bars indicate SEM of four replicate samples. Enzymes: α-glucosidase (AG), β-glucosidase (BG), β-D cellubiosidase (CB), Leucine amino peptidase (LAP), N-acetyl β glucosaminidase (NAG), Phosphatase (PHOS), and β-xylosidase (XYL).

communities to understand the overall system level response to plastic films, we likely missed changes happening on smaller spatial and temporal scales. For example, *Li et al. (2014b)* reported changes in microbial communities in soils that were sampled in close proximity to buried mulch films, indicating that microbial communities in the immediate vicinity of the films may be affected. Here we show that any local effects of mulch films are not detectable at a field scale, at least over a two-year period.

We did note significant differences in soil bacterial composition by location and season, which has been observed in other studies (*Li et al., 2014b*; *Moore-Kucera et al., 2014*). In our study, mulch effects were minimal compared to other drivers of community structure variation. It is well accepted that local soil conditions such as temperature, moisture

Table 4 *F* values obtained from a mixed model analysis of variance of the soil enzyme activities. Significant values are in bold, *$p < 0.05$; **$p < 0.01$; ***$p < 0.001$.

| Location | Factor | β-xylosidase | β-glucosidase | α-glucosidase | N-acetylβ glucosaminidase | β-D cellubiosidase | Phosphatase | Leucine amino peptidase |
|---|---|---|---|---|---|---|---|---|
| TN | Treatment | 2.21 | 1.49 | 2.62 | **2.53***  | 1.03 | 1.37 | 1.71 |
| | Time | **46.48***** | **29.56***** | **40.16***** | **34.60***** | **32.82***** | **68.23***** | **28.83***** |
| | Treatment * Time | 1.55 | 0.92 | 1.52 | 1.26 | 0.88 | 1.04 | 0.96 |
| WA | Treatment | 0.89 | 0.84 | 1.12 | 0.64 | 0.75 | 1.13 | 0.34 |
| | Time | **5.12***** | **3.44*** | **13.31***** | **6.06***** | 0.27 | **4.10**** | 0.65 |
| | Treatment * Time | 0.88 | 0.91 | 0.65 | 0.78 | 0.72 | 0.96 | 0.77 |

and pH play a pivotal role in shaping microbial communities (*Fierer & Jackson, 2006*; *Moore-Kucera et al., 2014*; *Rousk et al., 2010*). In this study, the location differences in communities were attributed to higher relative abundances of *Acidobacteria, Actinobacteria* and *Planctomycetes* in TN and higher abundances of β- and γ-*Proteobacteria* in WA. This corresponds with higher pH and saturated K in TN and higher soil organic matter and soil moisture in WA. Both pH and water content are major edaphic factors that influence temporal and spatial variation in soil microbial communities (*Docherty et al., 2015*; *Rousk et al., 2010*). Changes in soil physicochemical properties and different climates and soil types between TN and WA could explain such locational differences. Seasonal differences in communities were driven by significantly increased percent relative abundance of *Acidobacte*r *Gp6, Gp4* and *Gp7* in Spring in TN as compared to Fall. Additionally, significantly greater abundances of *Planctomycetaceae* and *Streptomyces* were seen in Fall compared to Spring in TN. In WA, *Acidobacteria_Gp6* and *Spartobacteria* showed significantly greater percent abundances in Spring compared to Fall whereas *Streptomyces* showed significantly higher percent abundance in Fall compared to Spring (Fig. S2). Seasonal tillage operations often reset many of the soil properties, which can explain why the abundances of some taxa oscillated between Spring and Fall. Actinobacteria such as *Streptomyces* have demonstrated polymer degrading capabilities (*Pathak & Navneet, 2017*). However, because we did not observe differences in the relative abundance of this taxa between BDMs, PE or no mulch control, this increase is likely attributable to the agronomic management of the plots (e.g., plant species, irrigation or fertilizer regimes etc.), rather than a response to mulch type.

Mulch materials did not have a consistent impact on bacterial richness or diversity. A previous study evaluating microbial diversity using PCR-DGGE showed no difference in ammonia oxidizer diversity under biodegradable and non-biodegradable mulching materials one year after tilling plastics into soil (*Kapanen et al., 2008*).The higher richness estimates under BDMs compared to PE treatments, which was significant in Fall 2015 in WA, suggested that tilled BDMs may help promote richness in the soil environment.

Using gene copy abundances as a proxy for bacterial and fungal abundances, we observed some evidence of a BDM-induced enrichment. In WA, both bacteria and fungi increased in abundance under BDM and Weedguard treatments over the course of the two-year experiment. Because we did not see an increase under PE, this suggests that this is in response to the incorporation of BDMs into the soil (as opposed to an indirect effect of microclimate modification, such as soil warming). In TN, we observed bacterial, but no fungal, enrichment in two of the four BDM plots and PE plot. Previous studies have also demonstrated increased fungal abundances in soil because of BDM incorporation (*Li et al., 2014b*; *Ma et al., 2016*; *Muroi et al., 2016*; *Rychter et al., 2006*). Fungi are important colonizers and degraders of BDMs; several plant pathogenic fungal species such as *Alternaria brassicicola Aspergillus fumigates, Humicola insolens,* and *Aspergillus oryzae* are known to produce cutinases which can accelerate degradation of biodegradable mulch films (*Koitabashi et al., 2012*; *Moore-Kucera et al., 2014*; *Muroi et al., 2016*). There is precedent for the differential responses in microbial enrichment we observed between the two locations, with both fungal and bacterial enrichment in WA, but only bacterial enrichment

in TN. In a similar study comparing BDM effects in three locations, it was found that BDMs resulted in soil fungal enrichment in Lubbock, TX, and bacterial enrichment in Knoxville, TN (*Li et al., 2014b*). From other soil systems, we know that soil pH can be the best predictor of bacterial community composition, while fungal communities were more closely associated with changes in soil nutrient status (*Lauber et al., 2008*). Both TN and WA soils had comparable fungal gene abundances initially (Spring 2015). However, since the microbial communities in WA were related to organic matter (Fig. S1) and WA soils had higher C:N ratios than TN soils this could explain the fungal enrichment in WA but not in TN.

Enzyme assays were conducted to assess potential activity rates for common carbon, nitrogen and phosphorus cycling enzymes in soil. As with bacterial community structure, enzyme activity profiles showed the greatest differences by location and season (Fig. 5, Table 4). The seasonal oscillation in enzyme activities seen for almost all the enzymes could be attributed to seasonal tillage operations which tend to offset many of the soil biological functions (*Alam et al., 2014*; *Busari et al., 2015*; *Zuber et al., 2015*) (Fig. 6). This was also observed for many of the soil physicochemical properties (*Sintim et al., 2019*). Mulch treatments had significant effects on N-acetyl- β-glucosaminidase (NAG) in TN. NAG was decreased under mulches compared to no mulch treatments, with the greatest decrease observed under PE. NAG catalyzes the hydrolysis of chitin oligomers to form amino sugars which are major sources of mineralizable nitrogen in soils and thus is important in carbon and nitrogen cycling in soils. Xylosidase activity was also reduced under mulch treatments compared to no mulch plots in TN though not significantly. Because we saw decreases under all mulch treatments for NAG in TN, this is likely an indirect effect of the mulches via microclimate modification, rather than a direct effect of mulch fragments tilled into the soil. All mulches warm the soil, with PE often having a greater soil warming potential compared to BDMs (*Kader et al., 2017*; *Moreno & Moreno, 2008*). Mulches also increase soil moisture levels (*Qin, Hu & Oenema, 2015*). Consequently, changes in soil temperature and moisture will affect enzyme pool sizes (*Steinweg et al., 2013*). The reduction in activity under plastic mulches may be because TN has a warmer climate where plastic mulches can push temperatures above optima limiting soil microbial activity (*Moreno & Moreno, 2008*). Mean soil temperatures in summer under mulched plots were 24.7 °C at 10 cm depth in TN, whereas in WA it was 18.7 °C. Un-mulched plots had mean summer soil temperatures of 23.8 °C for TN and 17.0 °C for WA (*Sintim et al., 2019*). In the month of June in both years, soil temperatures exceeded 30 °C under mulched plots in TN, but were less than 30 °C for no mulch plots. It has been reported that fungal and bacterial growth rates have optimal temperatures around 25 to 30 °C in agricultural and forest humus soils, while at higher temperatures lower growth rates are found (*Pietikainen, Pettersson & Baath, 2005*). This decrease in growth rate was shown to be more drastic for fungi than for bacteria, resulting in an increase in the ratio of bacterial to fungal growth rate at higher temperatures. Thus, the high temperatures under mulches in the summer in TN were above optimum growth conditions for soil microbes and may have reduced soil enzyme activities. Cold-adapted microorganisms, which are expected to be more prevalent at the WA site, tend to respond more efficiently to increased temperature than warm-adapted microbes (*Brzostek & Finzi,*

2011). The greatest relative temperature sensitivity of decomposition processes has been observed at low temperatures (*Kirschbaum, 1995*). Warming experiments have revealed reduced xylosidase activity in soils under medium-warmed plots compared to unwarmed plots (*Steinweg et al., 2013*). It has also been reported that warming induces decreases in the temperature sensitivity of β-xylosidase activity in the H horizon (*Souza et al., 2017*). One study reported greater increase of the relative temperature sensitivity of XYL and NAG (important for C cycling) at lower temperatures, compared to amino peptidase enzymes suggesting that temperature plays a pivotal role in regulating the use of substrates. Thus, the turnover of easily degradable C substrates (like glucose) is more sensitive to temperature than higher molecular compounds, at least for cold soils (*Koch, Tscherko & Kandeler, 2007*).

Looking specifically at studies which assessed soil enzyme activities after treatment with biodegradable plastic film, one field study reported that soil microbial biomass and β-glucosidase activity were most responsive to mulch incorporation; however that study did not have PE as a control, so it is unclear if this response was specific to BDMs or just related to plastic mulching generally (*Li et al., 2014a*). The cited study also focused on soils in close proximity to plastic, rather than bulk soil responses. Laboratory studies have shown increased esterase activity in soils during the degradation of poly(butylene succinate-co-adipate) (PBSA) (*Yamamoto-Tamura et al., 2015*), and increased microbial activity per a fluorescein diacetate hydrolysis test during the degradation of a variety of biodegradable polymers (*Barragán, Pelacho & Martin-Closas, 2016*). These studies provide insight into the potential of these enzymes in the degradation process of BDMs. Other studies that have looked at more general activity responses by microbes under plastic mulches (i.e., respiration) have reported mixed results: some have observed increases in activity under plastic mulches (*Chen et al., 2017*; *Mu, Fang & Liang, 2016*; *Mu et al., 2014*; *Zhang et al., 2015*), while others report decreased activities (*Moreno & Moreno, 2008*).

## CONCLUSIONS

Two years of biodegradable and PE mulch treatments in a vegetable agroecosystem in two locations revealed only minor effects on soil microbial communities and their functions. We previously showed that biodegradable mulches did not have a significant impact on a suite of soil quality parameters at these sites (*Sintim et al., 2019*). The investigation of the microbial communities from the same experiment corroborate these results showing that locational and seasonal variations are more important drivers of changes in soil health under BDM tillage operations than the type of mulch treatment at these field sites.

It should be noted that marginal but significant location-dependent effects of mulches were observed. For example, in WA, BDM incorporation caused a significant enrichment in both soil bacterial and fungal abundances, suggesting a direct response to BDM incorporation into soils; while only bacterial enrichment was observed in TN. We additionally observed decreases in specific enzyme activities (NAG) under mulch treatments in TN but not WA, which may be attributable to increased temperatures under the plastics (i.e., microclimate modification) in the warmer climate. Together, this shows that plastic

mulches had minor impacts on soil microbial communities and their functions. BDMs may have effects different from PE plastic mulches, and these responses may be location-specific. As microbes are the drivers of soil carbon and nutrient cycling, changes in bacterial and fungal abundances and/or activity can have repercussions for soil organic matter dynamics and nutrient availabilities. Longer term studies of repeated BDM incorporation are needed to determine if these microbial responses will significantly affect soil functioning and health. In addition, the fact that we saw different responses by the communities in two locations under identical management may mean that the ultimate impact of plastic mulching on soil may be dependent on local climate and soil conditions.

## ACKNOWLEDGEMENTS

Field experiments were designed and managed by A Wszelaki, C Miles, D Inglis and D Hayes, with help from staff at the East Tennessee Research and Education Center (Knoxville, TN) and Northwestern Washington Research and Extension Center (Mount Vernon, WA). We are grateful to BioBag Americas, Inc. (Palm Harbor, FL, USA), Organix Solutions (Maple Grove, MN, USA), Custom Bioplastics (Burlington, WA, USA), (Metabolix Inc., (Cambridge, MA, USA), and Sunshine Paper Co. (Aurora, CO, USA) for the donation of mulches for the research experiments, and Techmer PM (Clinton, TN, USA) for preparation of the carbon black dye masterbatch and compounding of the PLA/PHA formulation used to prepare the PLA/PHA mulch film. A Saxton provided statistical advice. We thank M English, J Moore, S Schexnayder, M Flury, M Valendia, S Schaeffer, M Anunciado, K Henderson, S Keenan, LS Taylor, J Liquet, S Ghimire, A Bary, M Starrett, D Cowan-Banker and other members of the JMD and M Flury labs for help with soil sample collection and processing. M Flury and D Hayes provided critical feedback on the manuscript.

### Funding

This work was supported by the United States Department of Agriculture Specialty Crops Research Initiative (Award 2014-51181-22382). The funders had no role in study design, data collection and analysis, decision to publish, or preparation of the manuscript.

### Grant Disclosures

The following grant information was disclosed by the authors:
United States Department of Agriculture Specialty Crops Research Initiative: 2014-51181-22382.

### Competing Interests

The authors declare there are no competing interests.

### Author Contributions

- Sreejata Bandopadhyay conceived and designed the experiments, performed the experiments, analyzed the data, prepared figures and/or tables, authored or reviewed drafts of the paper, and approved the final draft.

- Henry Y. Sintim performed the experiments, authored or reviewed drafts of the paper, and approved the final draft.
- Jennifer M. DeBruyn conceived and designed the experiments, performed the experiments, analyzed the data, authored or reviewed drafts of the paper, and approved the final draft.

### DNA Deposition

The following information was supplied regarding the deposition of DNA sequences:

Sequence reads are available in the NCBI sequence read archive: PRJNA564156.

### Data Availability

The following information was supplied regarding data availability: Mothur code, R code and associated input files are available at GitHub: https://github.com/jdebruyn/BDM-Microbiology.

### Supplemental Information

Supplemental information for this article can be found online at http://dx.doi.org/10.7717/peerj.9015#supplemental-information.

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
