# Peer review of "Effects of biodegradable plastic film mulching on soil microbial communities in two agroecosystems"

_PeerJ, doi:10.7717/peerj.9015_

## Round 0.1 · original submission · Major Revisions

Dear Dr. DeBruyn,

Your manuscript has been reviewed by two external reviewers, who found that your work has merit but also suffers from some deficiencies. Please prepare a revised version that will take into account the comments and suggestions they have made.

In particular, both reviewers raised issues about the experimental design (e.g. regarding the relevance of sampling dates). Please address this concern very seriously. See in particular the comments made by Reviewer 2.

More generally, you should refrain from over-generalizing the conclusions allowed from the study.

I look forward to receiving a revised version of your manuscript along with a point by point response to the reviewers' comments and suggestions.

best regards

Xavier

Reviewer 1 ·

Basic reporting

• Overall, the manuscript is well-written, with appropriate support from previous studies. The comparison of PE with BDM in the same study is critical for increasing understanding of the impact of these management systems on agroecosystems, as the authors state.
• Line 67: Could the authors provide a source for this statement? What regulatory agencies are involved in these decisions?
• Line 85-86: “….we compared the impacts on soil microbial communities of BDM and PE mulch in two-year….
• Line 88: “measurements of a suite of soil….”
• Line 388: typos?
• Line 439: “….this suggests that this is in response to the incorporation…”

Experimental design

• Line 147: I assume the May collection was with plastic and the September collection was after the plastic was removed or tilled? How long before plastic was applied were the samples collected? How long after plastic was tilled were samples collected? How far apart from each other where the 30 soil cores per plot?

Validity of the findings

• Line 444: Why are fungi important colonizers and degraders of BDMs? Is there any relation to changes in soil nutrient content? Carbon? The authors discuss carbon inputs are briefly at the beginning of the Discussion section, but it’s not clear how those inputs might be tied (if at all) into a potential impact/difference between BDMs and PE.
• The ambiguity in the collection times (see Experimental Design) made it a little difficult to follow what comparisons were made between locations. The authors indicate that some of the differences between their findings and that of another BDM study may have been the time of interaction between the soil microbes and mulch. Clarification on what this time period was for this study would also help put the authors results into a better context with this other study.

Additional comments

Again, this is a well-written study. However, in the introduction, greater clarification on the link between soil health and microbial communities may strengthen the overall research question. The authors indicate that a previous study found that the “effect on soil health was minimal” (Lines 88-91). As microbial communities are a component of soil health (though not necessarily “officially” measured as such), it would be good to make a clear distinction on the differences between the previous “soil health” measurements and those being addressed in this study.

Reviewer 2 ·

Basic reporting

The article meets the standards and all aspects included in this basic reported are satisfactory.
However, there are other relevant improvements to incorporate. These are included in the attached file.

Experimental design

The research is relevant and meaningful. The technical standards of the research carried out are high. However, the experimental design covers knowledge gaps only partially, see attached file.

Validity of the findings

Research standards are met; results are robust and statistically sound.
Revision of the paper and improvement is required due to the paper experimental design.
Some conclusions are not directly derived from the paper results and are to be refined.
See details in the attached file.

Annotated reviews are not available for download in order to protect the identity of reviewers who chose to remain anonymous.

---

## Round 0.2 · Minor Revisions

Dear authors,

Your manuscript has been much improved, which is recognized by the reviewers who have re-evaluated it. However, both reviewers raised a few remaining issues which still have to be addressed.

I hope you appreciate that the objective of this further iteration on your manuscript is to further improve it, and I look forward to receiving the revised version.

Best regards
Xavier LE ROUX

Reviewer 1 ·

Basic reporting

The authors have done a good job responding to previous comments.

A minor change could be made to Line 116: "...but generally not explicitly included in assessments of soil health."

Experimental design

The authors have adequately addressed previous comments.

Validity of the findings

The authors have adequately addressed previous comments.

Additional comments

The authors have adequately addressed previous comments.

Reviewer 2 ·

Basic reporting

The manuscript has been substantially improved in basically all aspects required. It is now well written and, especially, better focused on the specific studies carried out. Overgeneralization is now avoided and, where needed, conclusions have been refined.

Experimental design

The experimental design is now clear to the reader. Several little but substantial changes have contributed to make the paper focused on the specific results found.

The research carried out is well defined, relevant and meaningful.

The contribution of the paper to advancing knowledge on the impacts of BDMs on the agricultural soil ecosystem is valued.

Validity of the findings

Regarding the conclusions, limitations arising from the experimental design stay and somehow restrict the impact of the paper findings. However, the authors make a very balanced discussion.

Two aspects are still requiring to be reviewed and modified:
• L 334-337: “Post hoc testing showed that the richness estimates in TN significantly differed between 2015 and 2016. In WA, Fall 2015 differed in richness from the other time points. In TN, Fall 2016 diversity was significantly higher than other seasons. Diversity estimates were significantly lower in Spring than in the Fall seasons for WA.

The whole paragraph should be revised and limited to really relevant findings. If authors consider they are relevant and worth to be mentioned, they should be supported by the data shown. Table 3 (referred to in line 334) shows significant differences due to location, time and time interaction with location and with treatment, but none of the specific differences mentioned can be inferred. They are also not evidenced in figure 3.

• L 344-345: “It is important to note, however, that Weedguard completely disintegrated in TN after the field season (Ghimire et al. 2018b)”

The term “disintegration” is unclear. Ghimire et al. 2018b do not determine “disintegration” but “percentage of soil exposure”, and only through direct visual observation. Small mulch fragments could remain imperceptible to the human vision. This is relevant since it has been acknowledged plastic fragments under 1mm size to cause significant effects on the environment.
In addition, Ghimere et al. 2018b do not report complete disintegration of the Weedguard mulch in TN. On the contrary they report percentage of soil exposure after the mulch treatment and crop growing season to be strongly dependent on the year. For Weedguard paper, figure 2 of the paper shows 100% TN soil was exposed (no mulch seen to remain on the soil) in 2015, but in 2016 only ca. 25% of the TN soil was exposed (75% of the soil remained covered by the mulch).

---

## Round 0.3 · accepted · Accept

Dear authors,

I am pleased to inform you that, following the revision made based on the reviewers' comments, your manuscript is now acceptable for publication in PeerJ.

Best regards

Xavier LE ROUX